# TGFβ-1 Induced Cross-Linking of the Extracellular Matrix of Primary Human Dermal Fibroblasts

**DOI:** 10.3390/ijms22030984

**Published:** 2021-01-20

**Authors:** Mariya E. Semkova, J. Justin Hsuan

**Affiliations:** Institute for Liver and Digestive Health, Division of Medicine, University College London, Rowland Hill Street, London NW3 2PF, UK; j.hsuan@ucl.ac.uk

**Keywords:** transglutaminases, lysyl hydroxylase 2, lysyl-oxidase-like enzymes, dermal fibroblasts, extracellular matrix, transglutaminase cross-linking, TGFβ-1

## Abstract

Excessive cross-linking is a major factor in the resistance to the remodelling of the extracellular matrix (ECM) during fibrotic progression. The role of TGFβ signalling in impairing ECM remodelling has been demonstrated in various fibrotic models. We hypothesised that increased ECM cross-linking by TGFβ contributes to skin fibrosis in Systemic Sclerosis (SSc). Proteomics was used to identify cross-linking enzymes in the ECM of primary human dermal fibroblasts, and to compare their levels following treatment with TGFβ-1. A significant upregulation and enrichment of lysyl-oxidase-like 1, 2 and 4 and transglutaminase 2 were found. Western blotting confirmed the upregulation of lysyl hydroxylase 2 in the ECM. Increased transglutaminase activity in TGFβ-1 treated ECM was revealed from a cell-based assay. We employed a mass spectrometry-based method to identify alterations in the ECM cross-linking pattern caused by TGFβ-1. Cross-linking sites were identified in collagens I and V, fibrinogen and fibronectin. One cross-linking site in fibrinogen alpha was found only in TGFβ-treated samples. In conclusion, we have mapped novel cross-links between ECM proteins and demonstrated that activation of TGFβ signalling in cultured dermal fibroblasts upregulates multiple cross-linking enzymes in the ECM.

## 1. Introduction

Fibrotic diseases are a major health issue that is estimated to be responsible for 45% of the mortality in the Western world [1]. The high mortality rate linked to this type of disease is a result of high incidence, the lack of validated biomarkers for early diagnosis, the lack of understanding of the molecular mechanisms underlying disease pathophysiology, the variations in aetiology depending on the affected organ, and the lack of effective disease-modifying therapies [2]. The fibrotic process is characterised by replacement of normal tissue with non-functional fibrotic tissue [3]. This condition is caused by abnormalities in normal tissue repair mechanisms that are controlled by regulatory mechanisms under homeostatic conditions. Under pathologic conditions, however, these regulatory mechanisms are suppressed, causing the uncontrolled accumulation of fibrotic tissue. The molecular pathways causing fibrosis can be divided into common pathways that can be found in every type of fibrosis [4], and disease-specific pathways that are unique to the affected organ [5]. Current strategies for anti-fibrotic therapies focus on fundamental common pathways in order to find universal anti-fibrotic drugs that could be used to treat any affected organ [5]. Among these pathways are transforming growth factor β (TGFβ) signalling and various proinflammatory pathways such as the profibrotic type 2 immune response [6]. The major event causing fibrosis is the uncontrolled accumulation of extracellular matrix (ECM) [7,8] that disrupts normal organ architecture and can finally lead to organ failure [3,9]. This effect relies on a number of molecular pathways including increased production of ECM proteins [10,11], a disturbed balance between the production of matrix metalloproteinases (MMPs) and tissue inhibitors of metalloproteinases (TIMPs) [12,13], and alterations in the mechanical properties of the ECM (increased matrix stiffness) [14]. Another molecular change, which is thought to impair ECM remodelling, is an alteration in ECM protein cross-linking [15]. Cross-linking of the matrix is catalysed by specific enzymes, including the lysyl oxidases, lysyl hydroxylases and transglutaminases, which are upregulated and hyperactivated in fibrotic tissue [16,17,18,19]. Excessive cross-linking changes the properties of the ECM by rendering it resistant to proteolysis [20].

Fibrotic disorders occur in a wide spectrum of systemic and organ-specific diseases. The former include systemic sclerosis (SSc), sclerodermatous graft vs. host disease and nephrogenic systemic fibrosis, and the latter include pulmonary, liver, cardiac and kidney fibrosis [21]. SSc is a complex connective tissue disease, whose most common features are vascular injury, inflammation, and extensive fibrosis that most commonly affect the skin [9,12]. Dermal fibrosis is caused by the release of various inflammatory cytokines and growth factors including TGFβ [22], platelet-derived growth factor (PDGF) and connective-tissue growth factor (CTGF) [9]. Pro-fibrotic TGFβ signalling [23,24] induces the differentiation of fibroblasts into profibrotic myofibroblasts that are responsible for the overproduction of ECM proteins [25]. Other events activated by TGFβ in SSc include the overexpression of CTGF [26] and various cytokine receptors [27,28] that enhance the inflammatory response.

The TGFβ signalling cascade has also been linked with an increase in the level of the transglutaminase 2 (TG2) and lysyl oxidases (LOX) ECM cross-linking enzymes. For example, a study of TGFβ-1 stimulated human dermal fibroblasts has revealed upregulation of TG2 protein in whole cell lysates [29]. Protein overexpression was accompanied by an increase in the level of lysate transglutaminase activity and the formation of fibronectin polymers [30]. These data suggest that TGFβ-1 stimulates TG2 cross-linking activity, but as the assays were performed on whole cell lysates, it remains unclear whether any such changes occur within the ECM compartment. Similarly, ECM-localised changes in cross-linking enzyme levels are unclear in studies on LOX and LOXL enzymes. For example, upregulated LOX and LOXL-4 mRNA levels were found in an in vitro model of skin tissue as a result of TGFβ-1 treatment together with an enrichment of the corresponding pyridinoline cross-linked products [31]. Moreover, TGFβ treatment also increases the expression of lysyl oxidase like 2 (LOXL2) in hepatocellular carcinoma cell cultures [32], and lysyl hydroxylase 2b (LH2b) in human synovial osteoarthritic fibroblast cultures [33].

Taken together, these results suggest that TGFβ-1 signalling induces changes in the ECM of dermal fibroblasts that lead to the reported increases in cross-linking. However, this hypothesis has not yet been supported by a direct analysis of changes in matrisome composition or enzyme activity. To address this, we performed proteomic and Western blot experiments to compare the amounts of different cross-linking enzymes in ECM isolated from control and TGFβ-1 treated cultures of primary human dermal fibroblasts. An upregulation of different cross-linking enzymes from the LOX, lysyl hydroxylase (LH) and transglutaminase families as a result of TGFβ-1 stimulation was detected. In addition, a significant increase in extracellular transglutaminase activity was demonstrated by measuring the incorporation of an exogenous amine substrate. Finally, transglutaminase cross-linking sites in control and TGFβ-1 treated ECM were compared using mass spectrometry (MS) in order to identify any new sites formed in response to sustained cytokine stimulation.

## 2. Results

### 2.1. Upregulation of LOXL and TG2 Enzymes by TGFβ-1

To test our hypothesis that TGFβ-1 treatment of cultured dermal fibroblasts increases the level and activity of cross-linking enzymes within the ECM, we first selected a suitable primary cell type. We chose dermal fibroblasts isolated from a healthy 41-year-old female donor as this donor age matches the typical age of SSc patients and is close to the average age at diagnosis [34].

A second critical aspect of the experimental design was to estimate the duration of incubation with TGFβ-1 that would show upregulation of cross-linking enzymes in the ECM. While TG2 protein levels were raised after 1 day of TGFβ-1 stimulation in a whole cell lysate [29], increased secretion of these proteins, their incorporation into the ECM, and measurable increases in target modification may require more time. To address this question, we conducted a pilot proteomic analysis of isolated ECM to assess the abundance of different cross-linking proteins after 5, 10 and 20 days of TGFβ-1 treatment (*n* = 1). During each period the dermal fibroblast cultures remained 100% confluent with no apparent detachment of cells either with or without TGFβ treatment. The results showed increased levels of several different cross-linking enzymes in the ECM only after 20 days of TGFβ treatment (Figure A1 and Figure A2).

To explore the statistical significance of this result, we repeated the 20-day experiment using biological triplicates for each condition, and we analysed both unnormalised and normalised protein levels (described in Section 4.6). The normalised values provided information about the amount of a protein relative to the overall range of the intensity-based absolute quantification (iBAQ) values. This relative value was independent of the total amount of protein used for the measurement. Thus, while an increase in the unnormalised protein value could have arisen from an increase in total ECM protein, comparing the normalised values of individual proteins allowed us to assess the enrichment of specific proteins within the ECM.

There was significant upregulation and enrichment of TG2 and LOXL1 as a result of sustained TGFβ-1 stimulation (Figure 1a,b). Upregulation of both enzymes by TGFβ-1 was consistent with the higher number of tryptic peptides identified for these enzymes under pro-fibrotic compared to control conditions (Figure 1c). The absence of LOXL1 from one of the control replicates caused a high variance in this dataset, which explains why statistical significance was not observed for the unnormalised iBAQ values. This effect was diminished after normalisation due to the relatively low abundance of this enzyme compared to other ECM proteins. The other two cross-linking enzymes from the LOXL family (LOXL2 and 4) also showed an increase in level as they were only found in the TGFβ-1-treated replicates (Figure 1c). A one sample t-test analysis detected a significant difference of the unnormalised values for LOXL2 from the expected value of 0. However, the level of variation in the normalised data was too high to detect a significant increase. LOXL 4 did not show a significant increase either, as the enzyme was absent from one of the TGFβ-1 treated replicates. The complete list of ECM proteins found in this experiment is given in Table A1.

### 2.2. Upregulation of LH2 in the ECM of Dermal Fibroblasts

The proteomics experiment did not identify any members of the lysyl hydroxylase family. A possible reason might be their low abundance in the dermal fibroblast ECM as detection of very low abundance proteins in a non-targeted experiment is typically challenging. Ions with far lower signal intensities than other signals in an MS1 spectrum are usually not selected for MS2 analysis and the corresponding peptide sequences are not identified. We therefore used Western blot analysis for the detection of LH2. This LH isozyme was of a particular interest, as its level had been shown to be elevated in fibroblasts isolated from SSc patients [35] accompanied by an increased accumulation of the corresponding pyridinoline cross-link products [36,37]. An increase in pyridinoline cross-links is associated with impaired ECM degradation, as collagenous matrices containing higher levels of these cross-links are more resistant to matrix remodelling by MMPs [38]. Hence, the LH2 enzyme appears to play an important role in fibrotic progression in SSc. Although LH2 is typically found in the cytoplasm, it is also secreted to the extracellular space [39]; thus, we expected to detect LH2 in the ECM preparation (pellet) and the cell lysate (supernatant). Figure 2 depicts the Western blot result showing the LH2 enzyme was indeed detected in both fractions. No change was apparent in cell lysate LH2 levels after normalisation to the GAPDH signal (*p* = 0.13). In contrast, significant upregulation of LH2 was apparent in the ECM extract (*p* = 0.04). The presence of additional bands was probably due to nonspecific binding of the primary antibody.

### 2.3. Increased Transglutaminase Activity in the ECM upon TGFβ-1 Stimulation

Despite the clear enrichment of TG2 in the cell culture ECM, this was not sufficient evidence for an increase in the cross-linking activity of this enzyme as most extracellular TG2 is inactive. We therefore assayed transglutaminase activity using a protocol similar to that published by Wang et al. [40]. In this assay we measured incorporation of an exogenous TG2 substrate, biotin-cadaverine, into cell-culture ECM after 15 days of treatment with or without TGF-β1. This treatment time was selected as it was the longest time point before any cell death in 96-well plates was observed. Prior to the activity assay, TG2 upregulation by TGFβ-1 at this time point was confirmed in cell lysates (*p* = 0.02) and ECM extracts (*p* = 0.003) using Western blot analysis (Figure 3a). The TG2 activity assay quantified incorporation of biotin-cadaverine via subsequent incubation with streptavidin-horseradish peroxidase (HRP). As we were only interested in the TG2 activity within the ECM, we measured HRP levels after decellularisation. Cell lysates were removed carefully to minimise disruption of the ECM on the culture surface. As HRP levels proved to be too low to be quantified using a chromogenic substrate, we measured the chemiluminescence emitted by a luminol substrate. In addition to samples prepared from control and TGFβ-1-treated cultures, we included two additional controls to determine the specificity of the assay for TG2 activity. One (blank control) lacked the TG2 substrate, and the other (negative control) included EDTA, which inhibits TG2 activity. The aim of this assay was to subtract nonspecific signals, arising from background binding of the substrate or the HRP conjugate to the ECM and not from TG2 activity, from the specific signal arising from bona fide TG2 activity in the ECM. Any basal level of activity in the control could not be detected possibly due to the relatively low signal-to-noise ratios (Figure 3c,d). The mean signal intensity produced by control ECM was not significantly different to that of the negative control (two-tailed two-sample t-test, *p* = 0.09). Consequently, any TG2 activity in control cultures was too low to quantify. This finding is consistent with a previous report showing that the majority of extracellular TG2 is inactive in the ECM of WI-38 fibroblasts cultured under physiological conditions [41]. In contrast, a >10-fold higher signal was detected in the TGFβ-1-treated cell culture ECM compared to the control cell culture ECM (*p* = 0.001).

### 2.4. Identification of Transglutaminase Cross-Linking Sites in TGFβ-1 Stimulated Dermal Fibroblast ECM

As the above results confirmed there was increased transglutaminase abundance and activity in dermal fibroblast ECM in response to sustained TGFβ-1 stimulation, we hypothesised that this would have led to an increase in the level of pre-existing transglutaminase cross-linked sites and the appearance of new sites in ECM proteins. Understanding such differences in the ECM cross-linking pattern between the control and the fibrotic model could help explain how increased cross-linking leads to the observed resistance of a fibrotic ECM to remodelling. To identify transglutaminase cross-linked sites in cell culture ECM, the ECM fraction prepared from dermal fibroblast cultures treated with TGFβ-1 for 30 days was compared with the corresponding untreated control. For this comparative study between the control and pro-fibrotic condition, treatment was extended to 30 days, as a longer incubation time was thought to improve the likelihood of detecting a difference in cross-linking pattern. Cross-linking sites were identified using MS/MS analysis of tryptic digests of each ECM fraction. Searches of the resulting MS/MS raw data were then performed using MassMatrix software, which is designed to identify cross-linked peptides from a protein database based on their precursor mass and fragmentation spectrum [42]. A cross-link identified in at least two of the three replicates prepared for each condition was considered to be correct.

The search for cross-linked peptides against a whole human proteome database is a very challenging task that has not been resolved [43]. Therefore, we restricted our search to a database containing ECM proteins that are known to be TG2 targets, including collagen type I, III and V [44,45,46], fibrinogen [47] and fibronectin [48]. Minimum score thresholds were established based on typical MassMatrix scores obtained from a tryptic, transglutaminase cross-linked control peptide (manuscript submitted). This threshold was important for the selection of reliable identifications. Another criterion used for identifying reliable matches was the number of MS2 spectra found for a cross-linked peptide. This was set to two or higher, as we found using a model cross-linked peptide that false-positive identifications were typically based on only one MS2 spectrum.

The five identified cross-linked peptides and their corresponding parent proteins are listed in Table 1 and the MS2 spectrum of each peptide pair is shown in Figure 4. One modification found consistently in these searches was the deamidation of proximal, non-cross-linked glutamine residues to glutamic acid residues. According to previous studies, alongside their cross-linking activity, transglutaminases are also able to catalyse glutamine deamidation [49]. Thus, a deamidated glutamine proximal to a cross-link may provide additional evidence for the location of that cross-link.

A qualitative comparison of the cross-links identified in the TGFβ-1 treated and control replicates revealed that the cross-linked site in fibrinogen alpha was specific to the TGFβ-1 treated samples (Table 1), suggesting that identifiable levels of this cross-link existed only as a result of TGFβ-1 stimulation. All other cross-linked peptides were found in two or more of the control replicates.

Quantitatively, the detection frequency in replicates was consistently higher for all cross-links in TGFβ-treated compared to control samples. To more closely assess whether any of the cross-links present in the control and TGFβ-treated samples differed quantitatively, precursor ion intensities were normalised to the intensity of the ECM protein glia-derived nexin, which was selected because its intensity is not known to be affected by TGF-β. A comparison of the normalised values showed that only the collagen alpha-2(V) cross-link increased in response to TGFβ. The average values for the control and TGFβ-1 treated samples were 0.005 and 0.133, respectively. The corresponding MS2 spectra showed that the majority of the highly abundant peaks had been assigned to a fragment belonging to the identified cross-linked peptide (Figure 4), providing further evidence for a correct identification. However, significance testing was unreliable, as the peptide was found in only two replicates per condition.

As the peptide sequences (R/K)KK and (R/K)KGR can be found in many ECM proteins, the identity of the protein or proteins that contained these cross-linking sites could not be determined. However, amongst all ECM proteins, the (R/K)GKGR sequence is found in only three proteins: collagen alpha-3(V), where it occurs twice as a proximal repeat; the CUB and EGF-like domain-containing protein 3 (SCUBE3); and the latent transforming growth factor beta binding protein 2 (LTBP-2). All other cross-linked sequences were found to be unique to the protein database that we used for our search (Table 1).

## 3. Discussion

Fibrosis is a complex disorder whose molecular drivers may differ depending on the affected tissue [2]. Thus, in order to discover key pathogenic mechanisms, each fibrotic type must be studied individually. Similar to other fibrotic pathologies, more research is required to better understand the underlying molecular events that lead to skin fibrosis in SSc [12], which is crucial for the development of new effective anti-fibrotic strategies. In this study, we present direct evidence that TGFβ-1 treatment of dermal fibroblasts triggers changes in ECM composition and structure. Such findings promise to help elucidate the role of TGFβ-1 signalling in the impaired ECM turnover observed during progression of skin fibrosis in SSc.

Previous studies have shown that TGFβ-1 treatment leads to increased expression of different cross-linking enzymes by different cell types. For example, TGFβ-1 treatment of hepatocellular carcinoma cells and osteoarthritic fibroblasts leads to increased expression of specific cross-linking enzymes from the LOXL and LH families [32,33], while similar treatment of dermal fibroblasts increased the expression and activity of TG2 [29]. In addition, increased LOXL and LH enzyme activity in response to TGFβ-1 has been shown in an in vitro model of skin [31]. However, as these previous studies assayed total cell extracts, changes in the ECM levels and activities of cross-linking enzymes were not understood. Moreover, transglutaminases are known to catalyse a wide range of different reactions in addition to their ECM cross-linking activity [50]. Accordingly, there was hitherto no direct evidence that increased TGFβ-1 signalling leads to increased LOXL, LH or transglutaminase abundance and activity in the ECM.

In addition to studies in which upregulation of different LOXL isoforms was thought to contribute to fibrosis, reciprocal downregulation of LOXL isoforms can inhibit fibrosis. For example, downregulation of the LOXL1 enzyme led to reduction of collagen and elastin expression alongside decreased elastin cross-linking in the murine CCL_4_ model of liver fibrosis [51]. Moreover, inhibition of LOXL2 activity by a monoclonal antibody resulted in a significant drop in cross-linked collagen, alongside suppression of TGFβ signalling, fibroblast activation and production of growth factors and cytokines in the bleomycin model of lung fibrosis [52]. Furthermore, a LOXL4 knockdown in an in vitro model of skin fibrosis suppressed TGFβ-1 induced collagen aggregation and cross-linking [53]. Taken together, the results of these various studies suggest that treatment of our SSc cell culture model with TGFβ-1 would increase the expression of LOXL enzymes in a profibrotic response.

While we expected TGFβ-1 to stimulate total ECM biosynthesis, our proteomics results are consistent with the hypothesis that TGFβ-1 also triggers significant changes in ECM composition. This was particularly evident from the increase that we found in the normalised iBAQ values of TG2 and LOXL1, which suggested that enrichment of these enzymes had taken place in the ECM. A similar ECM enrichment might also be true for LH2, although LH2 normalisation was limited to loading equal amounts of ECM onto the SDS-PAGE gel.

TG2 is a transglutaminase isoform that plays a crucial role in the development of pulmonary and cardiac fibrosis [54,55], where it has been hypothesized to play a crucial role in inhibiting collagen turnover as a result of its collagen cross-linking activity [54]. In this regard, it has also been shown that increased TG2 activity leads to decreased collagen digestibility by MMP-1 [56]. The increased transglutaminase activity that we identified in TGFβ-1 treated ECM produced by primary dermal fibroblasts indicates that this process is also relevant to skin fibrosis. Although it is widely known that TG2 is secreted and becomes active mainly in the ECM [57], it is possible that other TG isoforms were also activated. In this regard, TG1 is known to be active in the ECM of healthy skin [57] and several ECM proteins have been identified as targets for this isoform including fibrinogen α and β, as well as collagen type XII [58]. However, lower protein levels of other TG isoforms compared to TG2 most probably precluded detection of these TG isoforms in our proteomic analysis.

Our results showing an upregulation of TG2 combined with an increase in enzyme activity led us to the hypothesis that significant changes in the cross-linking pattern could be found in the ECM of TGFβ-1 stimulated fibroblasts. Little is known about the patterns of cross-linking sites and how they are affected in response to pro-fibrotic agonists. Several transglutaminase cross-linking experiments have been performed on isolated proteins, for instance on collagen isolated from calf skin [59]. Although preliminary information about enzymatic preference for specific protein regions can be gathered in this way, the isolation procedure can lead to loss of binding partners and modification of protein conformation that can alter the relative rates of cross-linking at different sites in a subsequent transglutaminase cross-linking experiment. As a consequence, it is unclear whether or not the reported cross-linking sites are relevant in vivo. Since the biological significance of the corresponding discoveries is questionable, there was no available literature with which our identified cross-links could be compared.

In our study, the proteins contributing four of the five cross-linked lysine sites could not be identified, primarily because these short peptides were probably derived from basic regions.

There are two points worth noting. First, that, while we did not identify the collagen alpha-3(V) protein in the proteomics experiment, possibly due to a low abundance, the GKGR, KK and KGR lysine-donor peptides can be found in the same region of this protein (Figure 5). Further studies are needed to determine whether transglutaminase cross-linking activity is high in this region of collagen alpha-3(V) relative to other regions of this and other ECM proteins, at least as far as lysine residues are concerned.

The location of cross-linking sites in this basic, non-helical region of the collagen alpha-3(V) protein is consistent with a study of rhabdomyosarcoma cell cultures that detected TG2 cross-linking activity only in the non-helical region of collagen V [45]. Furthermore, a microbial transglutaminase activity assay using a peptide substrate library showed that neighboring amino acid residues can play a crucial role in the rate of cross-link formation [60]. Hydrophobic and basic amino acid residues showed a particularly strong and positive influence on the rate of cross-link formation. Although the selectivity of transglutaminases is not yet fully elucidated and while it is not possible to identify protein substrates based on amino acid sequence alone, the high incidence of basic residues in the collagen alpha-3(V) non-helical region is consistent with the current understanding of substrate exposure and sequence preference.

The second point worth noting is that, amongst the three ECM proteins from which the (R/K)GKGR sequence could be derived, only LTBP-2 was detected in the fibroblast ECM proteome (average normalised abundance of 0.67). Latent transforming growth factor beta binding protein 1 (LTBP-1) is known to be cross-linked by TG2 as a part of the TGFβ activation mechanism [61], but it is not known whether other isoforms such as LTBP-2 are also targets for TG2. If the LTBP-2 identification is correct, then our results suggest that it can be cross-linked to fibronectin and fibrinogen alpha (Table 1). Studies on human lung fibroblasts and primary osteoblasts have found that fibronectin associates with LTBP-1, 3 and 4 only in the initial days of cell culture; however, as the ECM matures during extended culturing time (>10 days), LTBPs no longer colocalise with fibronectin [62,63]. These studies did not investigate LTBP-2, although another study found that LTBP-2 assembles into fibrillar structures in human fetal lung fibroblasts, similar to LTBP-1 [64].

In regard to the cross-linked glutamine peptides, previous studies have demonstrated that fibronectin can be cross-linked to any of the collagen types included in our protein database [65,66], although the cross-linking sites were not identified. The ECM cross-linking sites between fibronectin and collagen I, and possibly collagen V, that we identified using MassMatrix are consistent with these earlier studies. A previous study has demonstrated the formation of high molecular weight polymers of fibrinogen and type III procollagen by transglutaminase treatment [67], and further work is needed to identify the protein cross-linked to fibrinogen alpha in our ECM preparation. As of yet, no evidence is presented about the biological function of the cross-links between these proteins or their contribution to fibrotic progression.

Regarding the MassMatrix search of TGFβ-1-treated fibroblasts, the cross-link that was found in fibrinogen alpha was specific to samples treated with TGFβ-1. However, the exposure of our model to TGFβ-1 was short compared to the time for disease in patients to progress from increased ECM to fibrosis. In addition, multiple additional factors potentially affect cross linking in human disease, including other profibrotic signalling pathways; ECM remodeling in vivo; and ECM composition, stiffness and structure in a 3D context. To more closely assess cross-linking changes in chronic disease, our approach can be extended to cross-linking analysis of models with considerably longer exposure to pro-fibrotic cytokines and a range of patients and healthy controls, which could lead to the discovery of additional cross-linking sites relevant to skin fibrosis. In addition, further work is needed to establish a reliable method for the quantitative comparison of cross-linking sites from MS-data. Such advances would open a way to identifying the accumulation during disease progression of the specific cross-linking sites that are primarily responsible for the inhibition of matrix remodelling.

## 4. Materials and Methods

All materials were purchased from Sigma unless otherwise stated.

### 4.1. Cell Culture

Primary dermal fibroblasts obtained with informed consent from a 41-year-old female donor (passages 2–5) were cultured in Dulbecco’s Modified Eagle Medium (Fisher Scientific U.K., Ltd., Loughborough, UK containing 10% FBS (Labtech International Ltd., East Sussex, UK), 100 U/mL penicillin, 0.1 mg/mL streptomycin. Cell cultures were allowed to reach 100% confluence, serum starved for 48 h, and then treated with 4 ng/mL TGFβ-1 (R&D Systems, Abingdon, UK) in serum-free medium. The medium was replaced with fresh medium containing TGFβ-1 every 2 days. Corresponding control cultures incubated without TGFβ-1 were prepared in parallel. For the proteomics and Western blot experiments, cells were grown in 75 cm^2^ flasks and TGFβ-1 stimulation was performed for a total of 20 and 30 days, respectively. Each sample was prepared using biological triplicates.

For transglutaminase activity assays, a suspension of the human dermal fibroblasts in medium containing 10% (*v/v*) FBS was pipetted into white 96-well, optical-bottom plates (Nunc MicroWellTM, Merck Chemicals Limited, Dorset, UK). Approximately 4000 cells were seeded in each well and grown to 100% confluence. After serum starvation for 72 h, the medium was replaced with either serum-free medium containing TGFβ-1 or serum-free medium and incubated for a further 15 days. Parallel cell cultures were grown in 25 cm^2^ cell flasks (conditions as above) for Western blotting experiments. Each sample was prepared using biological triplicates.

### 4.2. ECM Preparation, Digestion and Fractionation

The medium was removed from each cell culture flask, followed by four washes with 4 mL PBS buffer at 20 °C. Cell cultures were then shaken in 1 mL decellularisation buffer (25 mM Na_2_HPO_4_, 0.1 M Na_2_CO_3_, 1% (*w/v*) Tween 20, 0.5% (*w/v*) sodium deoxycholate, 20 mM EDTA, protease inhibitor cocktail (cOmplete™, Roche, Burgess Hill, UK) at 1:1000, pH 11) for 1.5 h at 20 °C and 160 rpm. The decellularised samples and supernatants were scraped from the flasks, transferred to tubes and centrifuged for 5 min at 12,000 rpm. Supernatants were carefully removed to fresh tubes, and the remaining pellets were washed three times with sample buffer (5 mM sodium acetate, 2.5 mM CaCl_2_, 25 mM NaCl, pH 6.8). Lipid and detergent extraction was performed by incubation in 200 µL 80% (*v/v*) ice-cold acetone at −20 °C overnight. Samples were then centrifuged (10 min, 15,000 × *g*) and the supernatants removed, followed by a second 80%-acetone extraction for 1 h at −20 °C. The pellet was then washed once with 100 µL MS-grade water and with 100 μL sample buffer.

Proteins were deglycosylated with chondroitinase ABC (0.05 U), heparitinase II (Amsbio, Abingdon, UK, 0.05 U) and keratanase (0.125 U) for 16 h at 37 °C and 1000 rpm. After the addition of 0.05 volumes of 200 mM ammonium hydrogen carbonate (pH 8), samples were incubated with PNGase F (New England Biolabs UK Ltd., Hitchin, UK, 1250 U) for 4 h at 37 °C and 1000 rpm. Reduction and alkylation were performed using 10 mM dithiothreitol for 2 h at 37 °C and 1000 rpm, followed by 25 mM iodoacetamide for 30 min at 20 °C in the dark. Proteins were digested using 1 μg Lys-C (Wako, Hampshire, UK) by incubation for 2 h at 37 °C and 1000 rpm, followed by addition of 3 μg trypsin (Promega UK Ltd., Southampton, UK) and incubation overnight at 37 °C and 1000 rpm, and then a further addition of 1.5 μg trypsin and incubation for 2 h at 37 °C and 1000 rpm.

The resulting peptides were fractionated using liquid-phase isoelectric focusing (Agilent 3100 OFFGEL Fractionator, Cheadle, UK). A pH gradient from 3 to 10 was employed, from which 12 fractions per replicate were obtained. OFFGEL fractionation was performed as described by the manufacturer using the following parameters: 4500 V, 200 mW, and 50 μA maximum current for 22 kVh. After focusing was complete, liquid-phase fractions were removed to clean tubes. Peptides remaining in each well were then extracted using 0.1% formic acid for 1 h and added to the corresponding tubes. Each fraction was purified using a stage tip (C18, binding capacity: 50 µg) [68] prior to LC-MS analysis. Purified fractions were dissolved in 25 μL 0.1% (*v/v*) formic acid followed by centrifugation for 3 min at 16,000 × *g*. Supernatants were carefully recovered to fresh tubes.

### 4.3. Mass Spectrometry

Not more than 2 µg sample, estimated using A280 (*ε* = 1), was used for each LC-MS/MS run. Using a NanoAcquity™ ultra performance LC (Waters, Elstree, UK), each sample was loaded onto a pre-column (NanoAcquity™ 10K 2G V/M trap column, C18, 5 µm, 180 µm × 20 mm) and peptides were separated using an inline separating column (Nikkyo Technos Co. Ltd., Tokyo, Japan, C18, 5 µm, 100 µm × 150 mm) at a flow rate of 0.4 µL/min. Separations were performed at 20 °C and the pressure range was 900–1000 psi. Peptide elution was achieved using the following gradient: 1% B (0–2 min), 1%–6% B (2–6 min), 6%–31% B (6–79 min), 31%–60% B (79–92 min), 60%–90% B (92–93 min), 90% B (93–98 min), 1% B (98–120 min). The composition of solvent A was aqueous 0.1% (*v/v*) formic acid and the composition of solvent B was 0.1% (*v/v*) formic acid in acetonitrile. The LC-MS/MS analysis of each set of replicates was followed by washing with the following solutions: 0.1% (*v/v*) trifluoroacetic acid (5 µL), 10% (*v/v*) formic acid (10 µL), 0.1% (*v/v*) trifluoroacetic acid (10 µL), 1% (*v/v*) NH3 in 50% (*v/v*) can (10 µL), 80% (*v/v*) methanol (10 µL), 80% (*v/can*ACN (10 µL) as described by Fang et al. [69]. Peptide identification was achieved using an inline LTQ Orbitrap Velos mass spectrometer (Thermo Fisher Scientific, Dartford, UK). The 20 most abundant precursor ions from each MS scan were isolated for MS/MS analysis and fragmented via collision-induced dissociation using helium gas. The *m*/*z* scan range was 400 to 2000 Da.

Mass spectrometry data were analysed using MaxQuant software (version 1.5.5.1) [70]. MS and MS/MS spectra were compared with theoretical spectra generated from tryptic in silico digests of protein sequences of the whole human proteome database (source: Uniprot, downloaded on 19.07.2016, proteome ID: UP000005640) with no more than two missed cleavages. The following amino acid modifications that may occur naturally or arise during sample preparation were included in the search: cysteine carbamidomethylation (fixed modification); and N-terminal carbamylation, asparagine deamidation, N-terminal glutamine cyclisation, methionine oxidation, proline oxidation, and N-terminal acetylation (variable modifications). Protein quantification was performed by calculating the iBAQ value of each protein, which is proportional to its molar amount [71].

ECM proteins were identified using the human matrisome database developed by Naba et al. [72] (downloaded in 2018). This part of the analysis was performed using Perseus software [73]. In brief, ECM proteins present in each list of protein identifications were identified and annotated based on their common protein and gene names in the matrisome database. Unannotated, non-matrisomal proteins were then eliminated from each list.

### 4.4. Western Blotting

Aliquots (300 μL) of the decellularised (ECM) sample and supernatant (cell lysate) scraped from each flask in the 15- and 20-day TGFβ-1 stimulation experiments and their corresponding controls (see previous section) were centrifuged for 5 min at 12,000 rpm. An amount of 200 µL of each supernatant (cell lysate) was transferred to a fresh tube and 500 µL H_2_O was added to the pellet (ECM) followed by vortexing and repeated centrifugation. The wash supernatants were aspirated and the washing procedure was repeated twice. After the last washing step, supernatants were aspirated until approximately 20 µL remained in each tube. An amount of 10 µL of 3 × SDS-PAGE sample buffer was added to give final concentrations of 2.5% SDS and 62.5 mM Tris-HCl (pH 7.2). Samples were mixed, incubated at 95 °C for 5 min, and centrifuged at 16,000× *g* for 3 min. The protein concentration in each ECM extract and cell lysate was estimated using the bicinchoninic acid (BCA) assay (Fisher Scientific, Loughborough, UK) by measuring the absorbance at 562 nm on a FLUOstar Omega plate reader (BMG LABTECH Ltd., Aylesbury, UK). Aliquots containing 2 µg of ECM protein and 13 µg of cell lysate protein were used for Western blot analysis.

Proteins were separated using SDS-PAGE (4%–15% TGXTM Precast Protein Gel, Bio-Rad Laboratories Ltd., Watford, UK) and transferred to a polyvinylidene fluoride (PVDF) membrane using an iBlot^®^ Dry Blotting System (Thermo Fisher Scientific, Dartford, UK) at 10 V for 9 min. Membranes were conditioned with Tris-buffered saline-Tween (TBST, 20 mM Tris-HCl, 150 mM NaCl, 0.1% Tween-20) buffer for 5 min, blocked with 5% milk powder in TBST for 1 h at room temperature, and washed three times with TBST (5 min each).

Membranes were probed with either anti-lysyl-hydroxylase-2 rabbit antibody (21214-1-AP, Proteintech, Manchester, UK) diluted 1:1000, or anti-TG2 rabbit antibody (15100-1-AP, Proteintech, Manchester, UK) diluted 1:500, in 5% milk powder in TBST at 4 °C overnight. Membranes were washed 3 times with TBST or 0.45 mM Tween-20 in PBS, and incubated with horseradish-peroxidase (HRP)-linked anti-rabbit-IgG antibody (1:2000, 7074, Cell Signalling Technology, Leiden, The Nederlands) in TBST or 0.45 mM Tween-20 in PBS. Membranes were washed again three times, followed by chemiluminescence detection using a luminol reagent (GE Healthcare UK Ltd., Littlechalfont, UK) and a FluorChemM detector (ProteinSimple, Oxford, UK).

To detect the loading control protein, membranes were then washed four times with TBST and incubated with stripping buffer (2% SDS, 62.5 mM Tris-HCl, 0.11 M 2-mercaptoethanol) for 30 min at 50 °C, followed by 6 washes with TBST and reblocking. Glyceraldehyde 3-phosphate dehydrogenase (GAPDH) detection was achieved using an anti-GAPDH rabbit antibody (10494-1-AP, Proteintech, Manchester, UK) at a dilution of 1:2000.

Quantification of blot images was achieved using ImageJ software [74]. Unnormalised signal intensities were used for the analysis of TG2 in ECM extracts, but all other values were first normalised to the corresponding GAPDH intensity.

### 4.5. Transglutaminase Activity Assay

Transglutaminase activity was quantified using a similar procedure to that described by Wang et al. [40]. The medium was removed from cell cultures and replaced with serum-free medium (blank); serum-free medium containing 10 mM EDTA (negative control); or serum-free medium containing 250 µM biotin-cadaverine (Thermo Fisher Scientific, Dartford, UK) and 1 mM DTT (control and TGFβ-1 stimulated samples). Each condition was measured using eight biological replicates. Cells were incubated at 37 °C for 2 h, and media were then replaced with 2 mM EDTA in PBS buffer. Cultures were decellularised with PBS buffer containing 0.1% (*w/v*) deoxycholate and 2 mM EDTA (5 min at room temperature with gentle rocking). The decellularisation buffer was removed and plates were washed three times with TBST (5 min, room temperature, with gentle rocking). Samples were then incubated with HRP-conjugated streptavidin (Thermo Fisher Scientific, Dartford, UK) diluted 1:1000 in TBST for 1 h at 37 °C. Plates were again washed three times with TBST, and detection was performed by adding a luminol reagent (GE Healthcare UK Ltd., Littlechalfont, UK). Chemiluminescence was detected at 440 nm (max. absorbance of luminol: 428 nm) using a FLUOstar Omega plate reader with a gain setting of 4000.

### 4.6. Statistical Analysis

The total (unnormalised) amount of each identified ECM protein was determined by calculating the iBAQ values using the MaxQuant software. Scale-to-interval normalisation was performed via linear transformation of the iBAQ values across the whole sample dataset, so that the lowest value in each dataset was normalised to 0 and the highest value to 1. To analyse the statistical significance of the normalised and unnormalised protein amounts between control and TGFβ-1 treated samples, a two-sample one-tailed t-test was used (*p* < 0.05). The same statistical test was performed to assess the significance in the Western blot and transglutaminase activity assay data.

### 4.7. Identification of Transglutaminase Cross-Linked Peptides

A search for transglutaminase cross-linked peptides within the LC-MS/MS data obtained from the 30-day TGFβ-1 treatment experiment was performed using MassMatrix software [42]. A human database was constructed containing the following ECM proteins: collagen alpha-1(I), collagen alpha-2(I), collagen alpha-1(III), collagen alpha-1(V), collagen alpha-2(V), collagen alpha-3(V), fibrinogen (α, β and γ) and fibronectin (Uniprot, downloaded 19.04.2019, accession numbers: P02452, P08123, P02461, P20908, P05997, P25940, P02671, P02675, P02679 and P02751, respectively). The allowed variable modifications were glutamine deamidation, methionine oxidation, proline oxidation; and the only allowed fixed modification was cysteine carbamidomethylation. The selected protease was trypsin with a maximum of two missed cleavages per peptide. A decoy search was performed using a database containing the reversed protein sequences. The minimum and maximum peptide lengths were set to 6 and 40 amino acids, respectively. The minimum output pp and pp2 values were 5 and the minimum pptag value was 1.3. The precursor and fragment mass tolerance were set to ±20 ppm and ±0.8 Da, respectively. Searches for transglutaminase cross-links were performed by manually adding to the software database a new cross-link between a lysine and a glutamine residue with loss of a water molecule (mass shift of −17.03 Da), corresponding to the transglutaminase cross-linking reaction. The search was performed using the Exploratory Search mode with the maximum number of cross-links per peptide set to 1. The results employed the inbuilt false discovery rate of 0%.

## 5. Conclusions

Our results suggest that TGFβ-1 signalling initiates changes in the ECM proteome of dermal fibroblasts that would lead to increased cross-linking. This information could help elucidate key mechanisms that trigger impaired ECM remodelling during the progression of skin fibrosis in SSc.

## Figures and Tables

**Figure 1 ijms-22-00984-f001:**
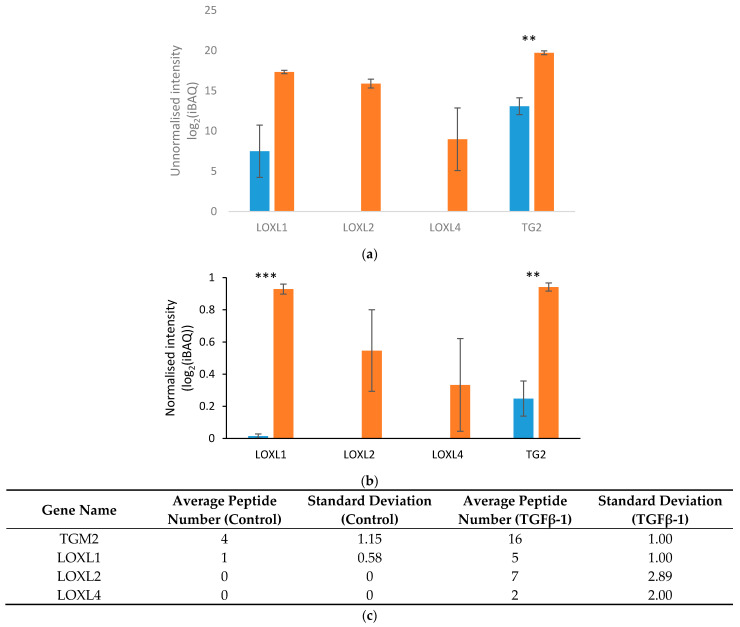
Proteomic analysis of cross-linking enzymes identified in dermal fibroblast ECM after 20 days of TGFβ-1 stimulation. A one-tailed, unpaired t-test was used to analyse the statistical significance of differences in protein abundance between control (blue bars) and TGFβ-1 treated (orange bars) cells (*n* = 3). (**a**) unnormalised iBAQ values (LOXL2: *p* = 0.002; TG2: *p* = 0.003); (**b**) normalised TG2 iBAQ values (LOXL1: *p* = 8.64 × 10^−5^; TG2: *p* = 0.003); (**c**) average peptide numbers identified for each cross-linking enzyme and the corresponding standard deviations in control and TGFβ-1-treated ECM preparations. Significant upregulations are marked with asterisks (two asterisks for *p* ≤ 0.01; four asterisks for *p* ≤ 0.0001).

**Figure 2 ijms-22-00984-f002:**
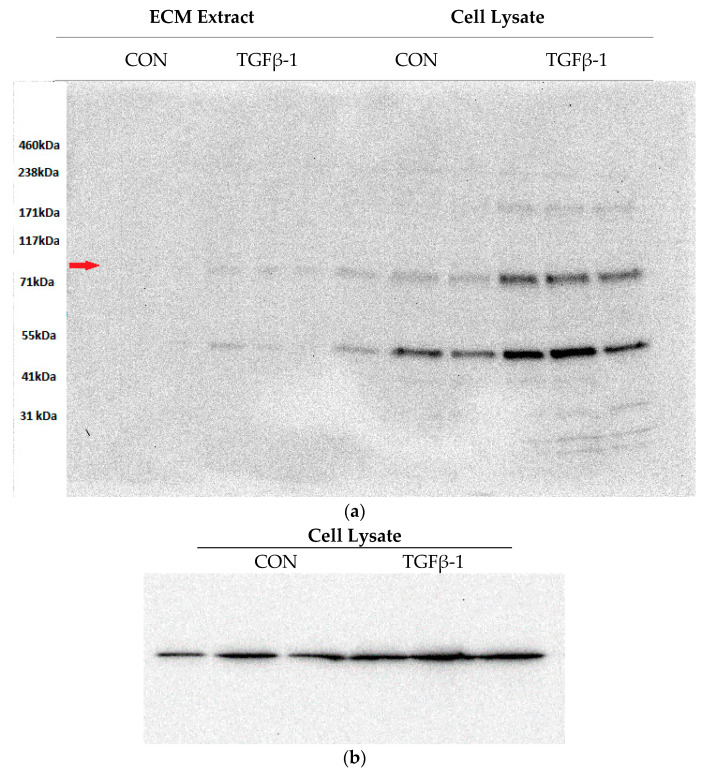
LH2 Western blot analysis of the 20-day stimulation of cultured dermal fibroblasts with TGFβ-1. ECM preparations from control or TGFβ-1-treated cultures, and cell lysate preparations from control and TGFβ-1-treated cultures were immunoblotted for LH2 (**a**), where the bands matching the expected enzyme molecular weight (85 kDa) are indicated with an arrow. The cell lysates were also immunoblotted for GAPDH (**b**); (*n* = 3).

**Figure 3 ijms-22-00984-f003:**
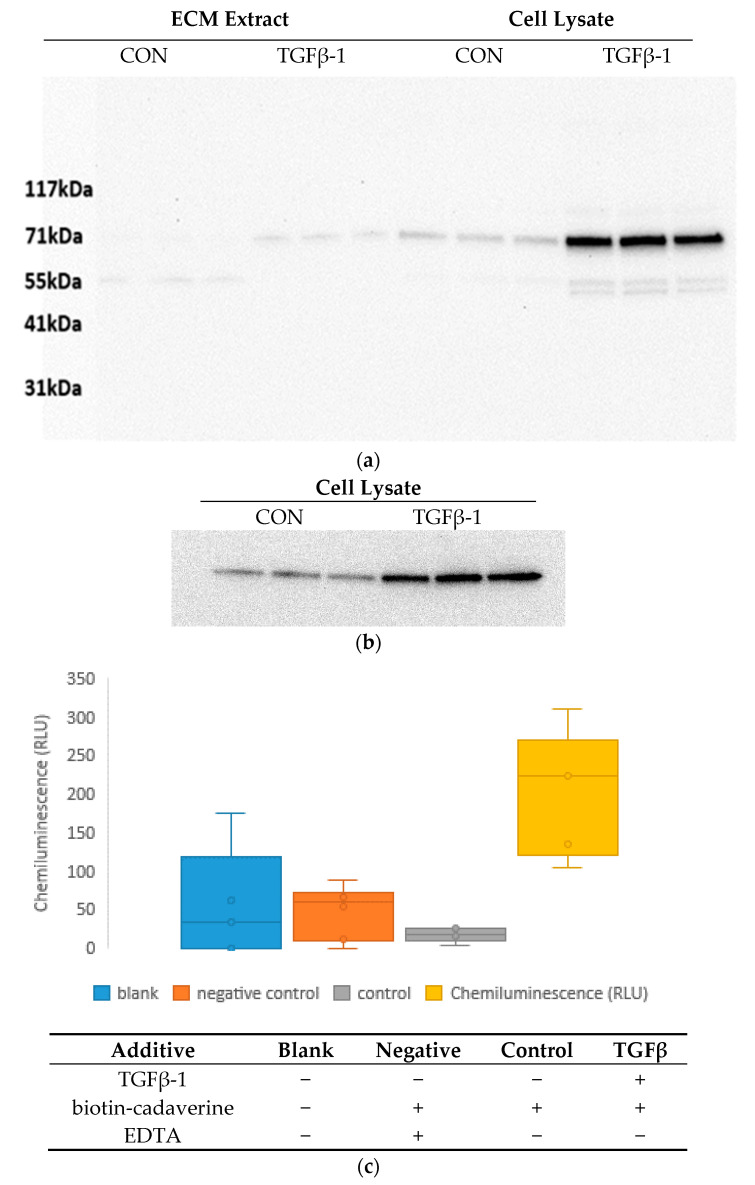
TG2 activity and Western blot analysis of decellularised control and TGFβ-1-treated cell cultures. ECM and cell lysates prepared from control and TGFβ-1-treated cultures after 15 days of incubation (*n* = 3) were immunoblotted with antibodies to TG2 (expected molecular weight 77 kDa) (**a**) and GAPDH (**b**). Panel (**c**) depicts the datasets (excluding outliers) in a box and whisker plot. Activity assay data are presented in relative luminescent units (RLU) alongside with information about the medium composition in each condition. The presence and absence of the additives are marked with + and—, respectively.

**Figure 4 ijms-22-00984-f004:**
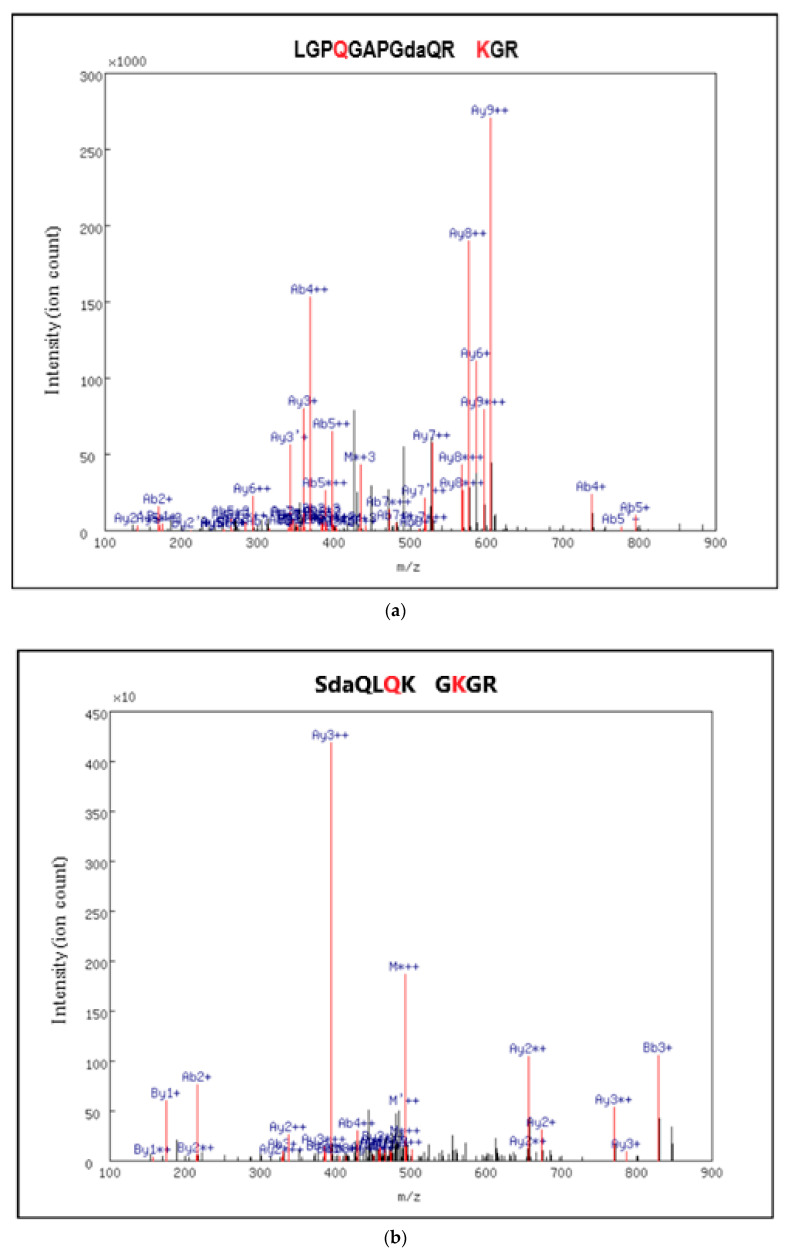
MS2 fragmentation spectra of the cross-linked peptides identified in the ECM of dermal fibroblast cultures treated with TGFβ-1 for 30 days. Cross-links between LGPQGAPGdaQR and KGR (**a**) SdaQLQK and GKGR (**b**) QGoxPSGASGER and VTPKEK (**c**) NNQK and GKGR (**d**) and GVQGoxPPGPAGPR and KK (**e**) were found. Cross-linked glutamine (Q) and lysine (K) residues identified by MassMatrix are shown in red text. Hydroxyproline and deamidated glutamine residues are labelled as oxP and daQ, respectively.

**Figure 5 ijms-22-00984-f005:**
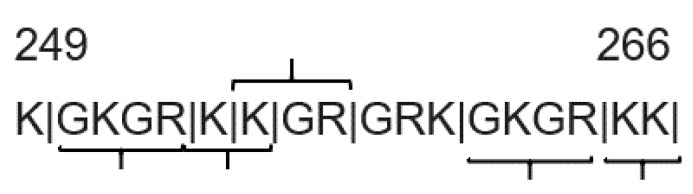
Amino acid sequence of the non-helical collagen alpha-3(V) region that contained multiple transglutaminase cross-links. Deduced trypsin cleavage sites are marked with vertical lines, and the sequences identified as cross-linked tryptic peptides by MassMatrix searches are marked with horizontal brackets.

**Table 1 ijms-22-00984-t001:** Cross-linked peptides identified in the ECM of dermal fibroblast cultures treated with TGFβ-1 for 30 days. Cross-linked glutamine (Q) and lysine (K) residues identified by MassMatrix are shown in red text. Hydroxyproline and deamidated glutamine residues are labelled as oxP and daQ, respectively. Alongside the peptide sequences, the number of replicates in which the peptides were found and the corresponding proteins are also given. The sequence number ranges are given in brackets after each parent protein.

Sequence + Modifications	Number of Positive Control vs. TGF Replicates	Cross-Linked Proteins
LGPQGAPGdaQR KGR	2/3 vs. 3/3	Collagen alpha-2(V) (347–366) + multiple candidates
SdaQLQK GKGR	0/3 vs. 2/3	Fibrinogen alpha (239–243) + three candidates
QGoxPSGASGER VTPKEK	2/3 vs. 3/3	Collagen alpha-1(I) (985–994) + Fibronectin (1847–1852)
NNQK GKGR	1/3 vs. 2/3	Fibronectin (2069–2072) + multiple candidates
GVQGoxPPGPAGPR KK	1/3 vs. 3/3	Collagen alpha-1(I) (686–697) + multiple candidates

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
