# Peer review of "TGFβ-1 Induced Cross-Linking of the Extracellular Matrix of Primary Human Dermal Fibroblasts"

_ijms, 2021, doi:10.3390/ijms22030984_

Round 1
Reviewer 1 Report
In this study, the authors hypothesized that increased ECM cross-linking by TGFβ contributes to skin fibrosis in Systemic Sclerosis (SSc) and they have mapped novel cross-links between ECM proteins and demonstrated that activation of TGFβ signaling in cultured dermal fibroblasts upregulates multiple cross-linking enzymes in the ECM and alters cross-linking between fibrinogen and collagen V.
Comments and suggestions:
In the introduction section, the authors should add more data about fibrosis disorders.
The section regarding used statistics is missing. The authors must add it.
In the Discussion section: please provide the limitations and clinical pitfalls of the study.
Consider revising accordingly.
Author Response
Thank you for your comments.
We have made the necessary changes as requested (see attached file).
- More information about the cause and relevant molecular pathways in fibrotic diseases has been added in the Introduction
- Section 4.6 Statistical Analysis has been added in the Materials and Methods
- The limitations of the study have been further discussed in the last paragraph of the Discussion

Reviewer 2 Report
The present manuscript sounds well written and novel.
The study is original and relevant in the field.
Methods are correct and rigorous. Tables and Figures are clear, useful and high qualitative. Findings interpretation and discussion are correct.
English is good and easy to read.
Author Response
Thank you for your comments.
Some changes have been made as requested by the first reviewer (see attached file).
- More information about the cause and relevant molecular pathways in fibrotic diseases have been added in the Introduction
- Section 4.6 Statistical analysis has been added in the Materials and Methods
- The limitations of the study have been further discussed in the last paragraph of the Discussion
